# Antioxidant Bilayers Based on PHBV and Plasticized Electrospun PLA-PHB Fibers Encapsulating Catechin

**DOI:** 10.3390/nano9030346

**Published:** 2019-03-03

**Authors:** Marina P. Arrieta, Alberto Díez García, Daniel López, Stefano Fiori, Laura Peponi

**Affiliations:** 1Institute of Polymer Science and Technology, ICTP-CSIC, Juan de la Cierva 3, 28006 Madrid, Spain; albertodiezgarcia1982@yahoo.es (A.D.G.); daniel.l.g@csic.es (D.L.); 2Departamento de Química Orgánica I, Facultad de Ciencias Químicas, Universidad Complutense de Madrid, Av. Complutense s/n, 28040 Madrid, Spain; 3Condensia Química S.A., R&D Department, C/La Cierva 8, 08184 Barcelona, Spain; s.fiori@condensia.com

**Keywords:** electrospinning, antioxidant active packaging, bio-based polymers, biodegradable polymers, poly(3-hydroxybutyrate-co-3-hydroxyvalerate) (PHBV), poly(lactic acid) (PLA), poly(3-hydroxybutyrate) (PHB), oligomeric lactic acid (OLA), catechin

## Abstract

The main objective of this work was to develop bio-based and biodegradable bilayer systems with antioxidant properties. The outer layer was based on a compression-molded poly(3-hydroxybutyrate-co-3-hydroxyvalerate) (PHBV)-based material while antioxidant electrospun fibers based on poly(lactic acid) (PLA) and poly(3-hydroxybutyrate) (PHB) blends formed the inner active layer. In particular, PLA was blended with 25 wt% of PHB to increase the crystallinity of the fibers and reduce the fiber defects. Moreover, in order to increase the stretchability and to facilitate the electrospinning process of the fiber mats, 15 wt% of oligomeric lactic acid was added as a plasticizer. This system was further loaded with 1 wt% and 3 wt% of catechin, a natural flavonoid with antioxidant activity, to obtain antioxidant-active mats for active food packaging applications. The obtained bilayer systems showed effective catechin release capacity into a fatty food simulant. While the released catechin showed antioxidant effectiveness. Finally, bilayer films showed appropriate disintegration in compost conditions in around three months. Thus, showing their potential as bio-based and biodegradable active packaging for fatty food products.

## 1. Introduction

There is a growing attitude towards the incorporation of active agents into food packaging materials instead of directly into the food, thus allowing a controlled release of the active component from the packaging to the foodstuff to maintain and even to enhance the food quality and safety [1,2,3]. Particularly, antioxidant packaging seeks to prevent the oxidation of food components (i.e., lipids and proteins), which lead to the deterioration of physical characteristics of food, such as flavor and color [4]. In fact, scavenging the primary free radicals of the oxidation process is the most efficient way to protect food against oxidation [1]. Catechin has been widely used for the development of antioxidant packaging materials mainly because it shows good scavenging activity [3,5]. Catechin is a natural flavonoid that can be obtained from several species of plants, in particular, from green tea and grapes [3,6,7]. The scientific literature reports many approaches for the development of antioxidant packaging systems, but most of them fail when they are trying to be developed into a packaging line at an industrial level [1]. In this sense, the electrospinning technique has gained considerable interest during the last year for the development of active layers for multilayer packaging systems [4,8,9]. Nowadays, the electrospinning process is a simple, extremely flexible, and low-cost process for fiber at an industrial level that can find several applications in the food packaging industry (i.e., active and intelligent packaging systems) [10,11]. It can produce multifunctional thin polymeric materials with different functionalities in the form of non-woven fibers from polymeric solutions subjected to high electric fields and at room temperature [2,12]. Food packaging is required to contain food products and protect them from the surroundings avoiding contamination, humidity, and oxidation processes [13,14]. Although electrospun fiber mats are not resistant enough to be directly used as films for food packaging, they can be used in multilayer packaging approaches [4,15,16]. Moreover, the electrospinning process is currently one of the most promising encapsulation techniques, and electrospun fibers have been considered interesting carriers for many active compounds [4,17,18]. Particularly, electrospun active layers have been recently developed and proposed as a food contact layer in multilayer systems [4,8,19].

On the other side, there is a current trend in the food packaging industry to substitute the use of non-renewable and non-degradable polymers with bio-based and biodegradable polymers. The use of bio-based polymers permits the reduction of the global dependence on fossil petrol sources for polymer production, while biodegradable polymers allow composting as a simple and sustainable end-life option for the packaging material. In this regard, catechin has been widely used in combination with biopolymeric matrices for the development of active packaging and composites [5,20,21,22,23]. Furthermore, in recent decades, bio-based and biodegradable polyesters, such as poly(lactic acid) (PLA) and the family of poly(hydroxybutyrates) (PHAs) have gained industrial attention. In this sense, the crystallinity of PLA has been successfully increased by blending with 25 wt% of poly(3-hydroxybutyrate) (PHB), allowing the formation of straight and bead-less electrospun fibers in comparison with neat PLA [10]. Moreover, catechin has been recently successfully incorporated into PLA-PHB (75:25) electrospun fiber mats obtaining fibers without defects and increased mechanical resistance when it was incorporated in amounts lower than 5 wt% [7]. To overcome the inherent brittleness of both biopolymers, plasticizers are frequently added to PLA-PHB blend systems [24], such as citrate esters [5,10], poly(ethylene glycol) (PEG) [25], and oligomeric lactic acid (OLA) [26]. Moreover, the combination of electrospun mats with a thermoplastic copolyester from the PHAs family such as poly(3-hydroxybutyrate-co-3-hydroxyvalerate) (PHBV) could represent an advantageous strategy since it will govern the resulting mechanical and barrier properties of the bilayer films [27]. In fact, PHBV has gained attention in the packaging field due to its commercial availability, ease of processing by using conventional thermoplastic equipment, and due to the fact that it possesses equilibrated mechanical properties in terms of stiffness and tensile strength [15]. Moreover, PHBV is frequently used to increase the barrier performance of multilayer packaging films [8,16]. However, PHBV still presents poorer mechanical performance than traditional packaging materials, such as lower impact strength and toughness [28]. In this sense, the modification of bio-based and biodegradable polymers by a blending approach has many advantages because it allows improving a wide range of the physical properties through cost-effective and readily available processing technologies at an industrial level [14]. It has been observed that by blending PHBV with an aliphatic-co-aromatic biodegradable polyester, poly(butylene adipate-co-terephthalate) (PBAT), a good balance of stiffness and toughness is obtained and, thus, PHBV applications are broadened [29].

In this work, PLA-PHB (75:25) electrospun fibers were loaded with 1 wt% and 3 wt% of catechin (Cat) to develop an antioxidant inner layer for biodegradable bilayer packaging systems. To facilitate the electrospinning process and to increase the stretchability of the final mats, 15 wt% of oligomeric lactic acid (OLA) was added as a plasticizer. Since electrospun mats are non-woven materials, an outer layer was required to achieve structural resistance of the final packaging material. To this end, a commercial PHBV-based pre-blend (PHBV/PBAT) was selected for the preparation of the outer layer by compression molding. The structural, thermal, and mechanical performance of each layer, as well as of the final bilayer system was studied. Since these bilayers are intended for active food packaging applications, the release ability of catechin, as well as its antioxidant response, was studied in a fatty food simulant. Finally, the disintegration under composting conditions at a laboratory-scale level of such bilayer systems was assayed to demonstrate their sustainable end-life option.

## 2. Materials and Methods

### 2.1. Materials

Poly(lactic acid) (PLA 3051, Mn = 110,000 Da, 3 wt% D-isomer) was supplied by NatureWorks (USA), poly(3-hydroxybutyrate) (PHB, under the trade name P226, Mw = 426,000 Da) was supplied by Biomer (Krailling, Germany) and poly(3-hydroxybutyrate-co-3-hydroxyvalerate) (PHBV, under the trade name ENMAT 5010P) was supplied by Tianan Biologic Materials Company, Ltd. (Ningbo, China) as a compound of PLA/PBAT in 45/55 proportion [30]. Oligomeric lactic acid (OLA 00A/8, Mn = 957 g mol^−1^) was synthesized according to a previously reported process [31] and kindly supplied by Condensia Química S.A (Barcelona, Spain). Catechin (Cat, 98% purity, anhydrous grade) was purchased from Sigma-Aldrich (Madrid, Spain). Chloroform (CL, 99.6% purity, boiling point 60 °C) and dimethylformamide (DMF, 99.5% purity, boiling point 153 °C) and 2,2-diphenyl-1-picrylhydrazyl (DPPH) 95% free radical were supplied by Sigma Aldrich (Madrid, Spain). 

### 2.2. Bilayer Systems Preparation

The inner layer was prepared by means of an electrospinning technique following previously optimized conditions for plasticized electrospun PLA-PHB-based materials [7,10]. Briefly, PLA pellets were previously dried at 80 °C overnight, PHB pellets, OLA, and catechin powder were dried at 40 °C for 4 h. Polymer solutions were prepared at 8 wt% in a mixture of cloroform:dimethylformamide (CL:DMF = 80:20 [32]) and further electrospun (polymer and solvent flow rate = 1.0 mL·h^−1^, positive and negative voltages = 10.8 kV and −10.8 kV and working distance = 14 cm) in a coaxial Electrospinner (Y flow 2.2.D-XXX, Nanotechnology Solutions). The electrospun fibers were randomly collected during 4 h and the obtained mats were vacuumed for 48 h to eliminate any potential residual solvents. Each formulation was prepared by blending PLA-PHB in 75:25 proportion on the basis of our previous results [10] and plasticized with 15 wt% of OLA. The plasticized PLA-PHB systems were further reinforced with 1 wt% and 3 wt% of Cat. To improve the Cat particles dispersion the solutions were sonicated during 10 min before being processed by electrospinning [7]. The obtained mat formulations and the proportion of each component are summarized in Table 1.

The outer PHBV-based layer was processed into films by compression molding at 180 °C in a hot press (Dr. COLLIN 200 × 200) by using a film mold (50 × 50 mm^2^). PHBV was previously dried at 40 °C for 4 h. PHBV pellets were kept between the plates at atmospheric pressure for 1 min until melting and they were further submitted to pressure cycle: 5 kPa for 1 min, 10 kPa for 1 min, and then quenched to room temperature at 5 kPa for 1 min. 

Finally, both layers were compression-molded to obtain a continuous bilayer film following already reported processes for the development of bilayer systems based on a PHBV outer layer with an electrospun inner layer [8]. In brief, a post-annealing process was applied by placing the inner electrospun layer onto the compression-molded PHBV and assembled in a hot press at 150 °C for 1 min and cooled down to room temperature in 2 min at 5 kPa. The obtained bilayer film formulations were labeled as the electrospun inner layer with the prefix PHBV: PHBV/PLA-PHB, PHBV/PLA-PHB-OLA, PHBV/PLA-PHB-Cat1, PHBV/PLA-PHB-OLA-Cat1, PHBV/PLA-PHB-Cat3, and PHBV/PLA-PHB-OLA-Cat3.

### 2.3. Characterization Techniques

The dynamic viscosity of the electrospun polymeric solutions was determined using an AR-G2 TA Instruments rheometer parallel plate geometry (40 mm in diameter). Rotational tests were conducted using a stepped shear rate from 0.01 to 1500 s^−1^ at 20 °C. 

The morphology of the obtained electrospun fibers in each mat and/or bilayer systems, as well as the cryo-fractured cross-sections of bilayer systems, were studied using a PHILIPS XL30 Scanning Electron Microscope (SEM). Samples were previously sputtered with a gold/palladium layer. The fiber diameters were statistically calculated from the SEM images with ImageJ software. 

Isothermal and dynamic thermogravimetric analysis (TGA) tests were conducted by means of a TA Instruments, TGA Q500 thermal analyzer. For isothermal TGA analysis, electrospun mats were heated at 150 °C during 30 min under air conditions. The electrospun mats, as well as the bilayer systems, were heated under TGA dynamic mode from 30 °C to 700 °C at 10 °C min^−1^ under a nitrogen atmosphere (flow rate 50 mL min^−1^). Film sample masses were between 5–7 mg. Initial degradation temperatures (T_0_) were determined at 5% of mass loss and the maximum degradation temperatures (T_max_) were calculated from the first derivative of the TGA curves (DTG).

The mechanical properties of monolayer and bilayer films were evaluated with the use of tensile test measurements conducted at room temperature by an Instron dynamometer (model 3366) equipped with a 100-N load cell, at a crosshead speed of 10 mm·min^−1^ and initial length of 30 mm. Dogbone-style film samples were used and at least five specimens were tested for each formulation. Release studies were performed in triplicate by the determination of catechin-specific migration tests into solutions of 50% v/v ethanol (food simulant D1) since these materials are intended for fatty food-packaging applications [33]. Thus, pre-weighed bilayer film samples were immersed in the food simulant D1 and were kept at 40 °C for 10 days. The released amount of catechin into the food simulant after 10 contact days was determined by the measurement of UV absorbance at 280 nm attributed to the B ring of catechin moiety, by means of a UV-Vis Perkin Elmer (Lambda 35, Waltham, MA, USA) UV-VIS spectrophotometer.

The antioxidant effectiveness of the released catechin was measured according to the DPPH-method [34] by determining the absorbance at 517 nm of the released catechin in the food simulant D1 at 10 days by means of a UV-Vis Perkin Elmer (Lambda 35, Waltham, MA, USA) spectrophotometer. The antioxidant activity was determined according to Equation (1):(1)$$\left(\%\right)=\left(\frac{A_{control}-A_{sample}}{A_{control}}\right)\times 100\%$$
where *I* (%) is the percentage of inhibition. *A_control_* the absorbance of DPPH at 517 nm in ethanolic solution and *A_sample_* is the absorbance of DPPH at 517 nm after 15 min in contact with the food simulant containing the released catechin. The % of inhibition was expressed as the equivalent of gallic acid (GA) concentration (mg kg^−1^) by using a calibrated curve of gallic acid concentration versus *I* (%).

The bilayer films were disintegrated in composting conditions at a laboratory-scale level following the ISO 20200 standard [35]. Samples (15 mm × 15 mm) were buried at a depth of 4–6 cm in appropriate reactors containing a solid synthetic wet waste (10% of compost (Mantillo, Spain), 30% rabbit food, 10% starch, 5% sugar, 1% urea, 4% corn oil, and 40% sawdust) and approximately 50 wt% of water content. The reactors were incubated under aerobic conditions at 58 °C. Film samples were recovered at different disintegration times (6, 23, 37, 51, 65, and 90 days). The disintegration degree at different days of incubation under compositing conditions was calculated by normalizing the sample weight to the initial weight.

## 3. Results

It is widely known that to prepare solvent-based electrospun biopolymeric fibers, the polymeric matrix should be homogeneously dissolved in a proper solvent [2,10]. In this context, good solubility of PLA and PHB matrices has been observed in CF:DMF in a proportion of 80:20 [7,10,36], and it has been ascribed to a similarity in their chemical structure which leads to similar solubility parameters (δ) [25] considering their group contribution according to the Small’s cohesive energies [37]: δ _PLA_ = 19.5–20.5 MPa^1/2^ [38] and δ _PHB_ = 18.5–20.1 MPa^1/2^ [39], while solvents also show similar solubility parameters: δ _CF_ = 19 MPa^1/2^ and δ _DMF_ = 24.9 MPa^1/2^ [32]. CF (boiling point 60°C) has been considered as an effective solvent for PLA-PHB blends; while DMF with lower evaporation rate (boiling point 153 °C) usually produces better electrospun PLA-PHB fibers [10]. Thus, allowing the solvent evaporation during electrospun fibers processing [7,10,32,36]. The good miscibility of PLA with OLA [40] has also been directly related with the high similarity in their chemical structure and solubility parameters (δ _PLA_ = 19.5–20.5 MPa^1/2^ [38] and δ _OLA_ = 17.7 MPa^1/2^ [41]). Catechin also shows the solubility parameter in the same order of magnitude (δ _Cat_= 11.9 MPa^1/2^ [42]) and if it is used as additive for PLA-PHB electrospun fibers in an amount less than 5 wt%, bead-free electrospun fibers can be obtained, while it induces a decrease of the average fiber diameter [7]. Therefore, the combination of PLA-PHB polymeric matrices in the proportion 75:25, the plasticization of the PLA-PHB blends in 15 wt% with OLA, as well as the use of catechin as antioxidant additive, in the amount of less than 5 wt% (i.e., 1 and 3 wt%) should be miscible.

The viscosity of the polymeric solutions was reduced with the presence of OLA plasticizer, see Table 1, but without the formation of droplets showing enough viscosity to be spinnable. Meanwhile, the viscosity of the polymeric solutions increased with the addition of catechin with a consequent higher average fiber diameter, see Table 1.

The morphological aspects, as well as the average fiber diameter of the electrospun mats, were studied by SEM, see Figure 1. A PLA-PHB mat exhibits randomly oriented uniform, straight, and bead-less electrospun fibers in agreement with already reported electrospun PLA-PHB (75:25) mats [10,36]. The plasticization of the PLA-PHB system with OLA lead to less straight fibers with a coarser surface and a slightly increased average fiber diameter, see Table 1. Cat increased the fiber diameter of PLA-PHB mats, see Figure 1c,e. Nevertheless, in plasticized systems Cat slightly reduced the fiber diameter, see Table 1. This behavior has been ascribed to the better interaction due to the presence of plasticizer which allows a homogenous distribution of Cat into the polymeric matrix which better interacts with all components in the system (PLA, PHB, and OLA) by means of hydrogen bonding [7]. Catechin-loaded plasticized electrospun PLA-PHB-OLA fibers show some spindle-like defects (beads), probably due to the reduced viscosity of plasticized electrospun solutions.

Figure 1g,h shows the cross-section of the bilayer PHBV/PLA-PHB and PHBV/PLA-PHB-OLA films respectively, as an example. The two layers with good adhesion between them can be clearly distinguished, the electrospun inner layer was very thin (thickness of about 20–30 µm) in comparison to the overall thickness of the bilayer system (200 ± 20 µm). The absence of phase separations between two polymeric layers has been ascribed to the thermal treatment between them [9]. However, less adhesion can be observed in the case of PHBV/PLA-PHB, see Figure 1g. Meanwhile, the presence of plasticizer in PLA-PHB electrospun fibers favors the adhesion of the mat to the PHBV surface, see Figure 1h. Moreover, in order to characterize the morphology of the electrospun fibers obtained after the hot-pressed assembly process used for bilayer preparation, the inner layer of the bilayer systems was observed by SEM. As an example, in Figure 1i,j the surface of the inner layer of PHBV/PLA-PHB-OLA of bilayer system is shown. It can be observed that the electrospun fiber structure was mainly maintained after heat compression treatment. However, fibers in direct contact with the hot plates in the compression molding press resulted in higher diameters and they showed less straight and coarser fibers. Thus, the overall average fiber diameter increased from 260 ± 78 nm in PLA-PHB electrospun fiber mats to 342 ± 120 nm in PHBV/PLA-PHB-OLA, showing more scattered values.

The effect of Cat and OLA on the thermal properties of the electrospun PLA-PHB blends was investigated by thermogravimetric isothermal and dynamic measurements. The isothermal TGA analysis, Figure 2, was carried out at 150 °C in order to ensure sufficient thermal stability of the electrospun active layer for assembly with the PHBV-based outer layer by compression molding for industrial purposes. Under isothermal conditions, no significant differences were observed between the PHBV-based pellet and the corresponding thermally processed PHBV-based films, suggesting that no thermal degradation has occurred during processing. Electrospun PLA-PHB mat showed slightly less thermal stability than the PHBV-based layer, which was slightly improved with 3 wt% of catechin addition (PLA-PHB-Cat3). The plasticized electrospun PLA-PHB mat (PLA-PHB-OLA) showed the lowest thermal stability. Meanwhile, plasticized systems loaded with catechin showed somewhat higher thermal stability since catechin protects the polymeric matrix from thermal degradation. It should be highlighted that the two layers are assembled into the final bilayer system by compression molding at 150 °C during 1 min and cooled down in 2 min. Thus, considering the actual assembly time of 1 min at 150 °C, the electrospun mats lost less than 0.5% of the mass.

The main thermal parameters obtained from the dynamic TGA and DTG curves, see Figure 3, are summarized in Table 2. Electrospun PLA-PHB and plasticized electrospun PLA-PHB mats present a two-step thermal degradation process in which PHB shows its maximum degradation temperature at around 280 °C (T_max1_) and PLA at around 340 °C (T_max2_). The presence of OLA plasticizer decreased the thermal stability of the electrospun PLA-PHB mats, shifting the onset degradation temperature to around 30 °C. Nevertheless, OLA improved the dispersion of catechin, since those materials with catechin at 3 wt% and OLA (PLA-PHB-OLA-Cat3) showed improved thermal stability with respect to the un-plasticized systems. It should be mentioned that the onset degradation of electrospun inner layers occurred between 197 °C and 258 °C, which is a temperature higher than that required for the compression-molded assembly process of bilayer systems, that is 150 °C, as it was already discussed for isothermal TGA results.

Catechin addition in 1 wt% increased the T_max1_ of PHB and T_max2_ of PLA (around 10 °C) in electrospun PLA-PHB-OLA-Cat1 with respect to the PLA-PHB-OLA counterpart, showing an effective stabilizing effect for both polymeric matrices. A higher amount of catechin (3 wt%) was not able to stabilize either the PHB or PLA matrix, as indicated by a significant decrease of the T_max1_ and T_max2_. This unexpected result has been related to the fact that the optimum stabilization effect of phenolic compounds on the polymer matrices is characterized by an optimum amount, while higher amounts over the optimal point do not provide a higher stabilization effect [43,44].

The PHBV sample showed a two-step degradation behavior since it is a compound of PHBV/PBAT, see Figure 3c,d and Table 3. The first step corresponds to the degradation of PHBV, while the second is the degradation of PBAT [30]. In bilayer systems, it was observed that, in general, all mats slightly reduce the thermal stability of PHBV, as it can be seen by the decrease of the onset of thermal degradation, as well as the maximum degradation temperatures. Nevertheless, those systems with higher amounts of catechin in the inner layer showed an increase of the T_0_ and the T_max1_ in bilayer films, see Table 3, with respect to their unloaded counterparts. The antioxidant ability of catechin protects the PHBV-based polymer compound from thermal degradation at the first stage [5]. Meanwhile, the T_max2_ of PBAT was shifted to lower values.

Figure 4 shows the mechanical properties of the monolayer (left) and bilayer systems (right). In monolayer systems, it is possible to observe that catechin acts as a filler, reinforcing the PLA-PHB matrix and leading to an increase of the Young’s modulus, as shown in Figure 4a, and tensile strength, see Figure 4c. The reinforcing effect of catechin particles in PLA-based materials has already been observed, particularly in low amounts when it is well dispersed into the polymeric matrix and is attributed to the strong interaction of catechin hydroxyl groups with PLA carbonyl groups [5,7,43]. In this work, this reinforcing effect was particularly marked when catechin was at 1 wt%, due to the good dispersion achieved at this proportion, as well as to the presence of fibers without defects (such as beads). On the other side, the presence of OLA increased the elongation at break, showing the success of the plasticizing effect, see Figure 4e. [40]. Regarding the bilayer systems, the fibers increased the Young’s modulus of PHBV-based monolayer, see Figure 4b, confirming the good adhesion in the bilayer structure. Meanwhile, the tensile strength, see Figure 4d, and the elongation at break, see Figure 4f, of the PHBV-based material were not significantly altered by the addition of the electrospun active layer.

The release of catechin from the bilayer systems was studied in fatty food simulant D1 and it was expressed as the amount of catechin and epicatechin released from the inner layer to the food simulant D1 at 1, 6, 10, and 20 days, see Figure 5a. Catechin presents high solubility in ethanol (50 g L^−1^), this is the reason why it is able to interact with fatty food simulant and be released from the polymeric matrix. The incorporation of OLA showed a significant increase in the catechin release capacity from electrospun PLA-PHB electrospun fibers. This behavior has been ascribed to the increased polymer chain mobility due to the plasticizing effect which favors the active compound release [3,5,18,26].

In order to confirm the effectiveness of the developed bilayer structures as antioxidant bilayer packaging systems, the reduction of stable free radical DPPH caused by catechin presence in the food simulant D1 was studied, see Figure 5b. A higher antioxidant effect was observed in the plasticized systems with a higher amount of catechin (3 wt%). This can be explained as plasticizer presence improved the release capacity of catechin from the polymeric matrix to the foodstuff and, thus, OLA leads to materials with higher antioxidant effectiveness than unplasticized ones. As expected, the behavior of the antioxidant activity as a function of time is comparable with the catechin release.

Finally, the disintegrability under composting conditions was conducted to corroborate the biodisintegrable character of the developed bilayer materials, see Figure 6. Although the biodegradable character was governed by the outer layer of PHBV, which is thicker than the electrospun layer, some insights regarding the influence of the electrospun layer can be obtained. After 23 days of disintegration, the bilayer systems started to become breakable, see Figure 6a, suggesting that the hydrolysis process starts in the electrospun materials and continues in the PHBV matrix. Thus, the electrospun layer somewhat speeds up the disintegration process. Catechin slightly delays the disintegration process, while OLA speeds it up. The disintegration process particularly increases when both additives were incorporated into the system. It is worth noting that all formulations were totally disintegrated under composting conditions in less than three months (90% of disintegration according to ISO 20200, Figure 6b), showing their inherent biodegradable character by thermophilic bacteria.

## 4. Conclusions

Biodegradable antioxidant bilayer films were successfully developed by using an outer layer based on a compression-molded PHVB/PBAT-based material and an inner antioxidant layer based on plasticized PLA-PHB electrospun fibers loaded with catechin. Plasticized electrospun PLA-PHB fibers encapsulating catechin were successfully obtained by means of the electrospinning technique. The addition of 1 wt% of Cat was enough to improve the mechanical properties of the electrospun mats, while 3 wt% produced some structural defects reducing the mechanical performance of the final materials. The obtained mats were used as inner layers for PHBV-based bilayer systems, where a PHBV/PBAT-based material provided the mechanical resistance to the final packaging film. The bilayer systems presented antioxidant activity in a fatty food simulant showing their potential as active food packaging materials. Meanwhile, they were totally disintegrated under composting conditions highlighting their potential application in the sustainable food packaging industry.

## Figures and Tables

**Figure 1 nanomaterials-09-00346-f001:**
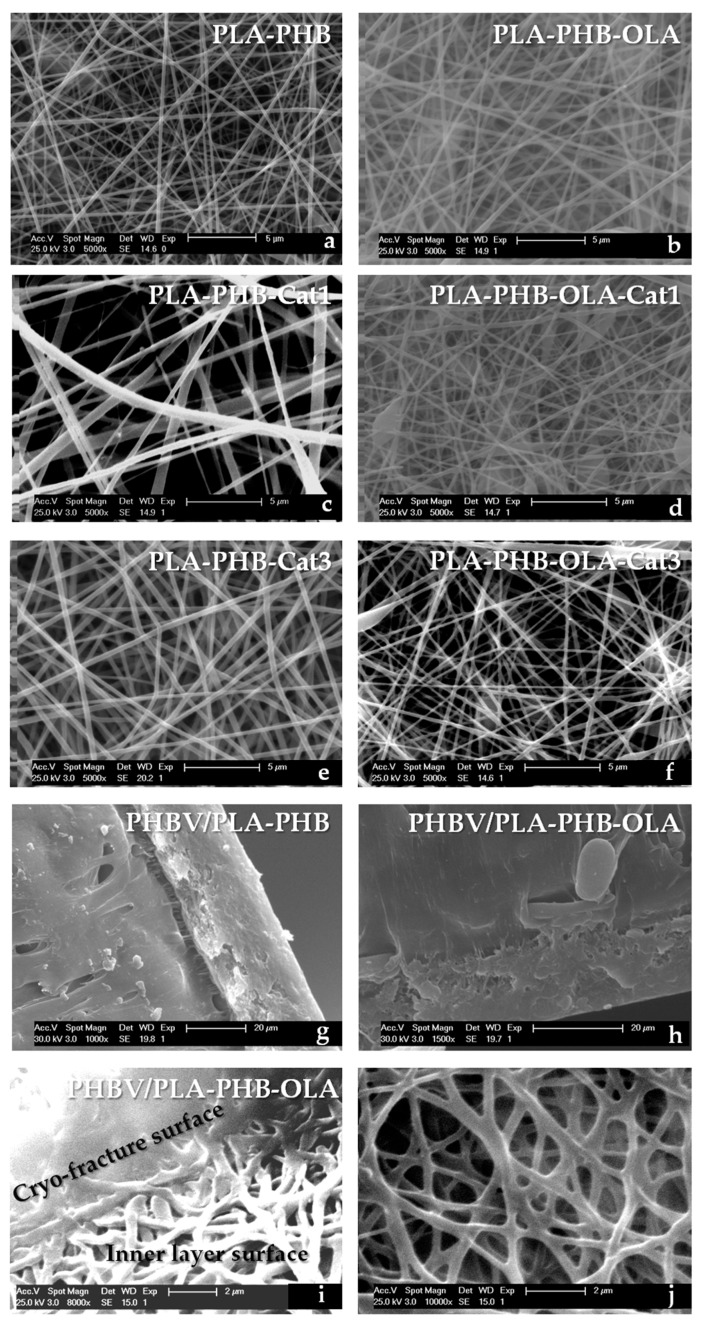
Scanning electron microscope (SEM) observations of electrospun mats: (**a**) PLA-PHB, (**b**) PLA-PHB-OLA, (**c**) PLA-PHB-Cat1, (**d**) PLA-PHB-OLA-Cat1, (**e**) PLA-PHB-Cat3, and (**f**) PLA-PHB-OLA-Cat3; SEM observations of the cry-fracture section of (**g**) PHBV/PLA-PHB and (**h**) PHBV/PLA-PHB-OLA as well as (**i**,**j**) inner surface of PHBV/PLA-PHB-OLA.

**Figure 2 nanomaterials-09-00346-f002:**
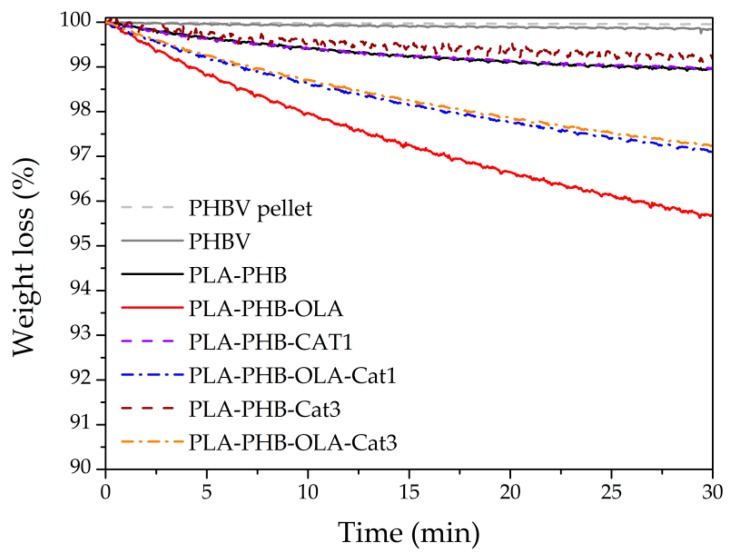
Isotheral thermogravimetric analysis (TGA) at 150 °C of electrospun mats.

**Figure 3 nanomaterials-09-00346-f003:**
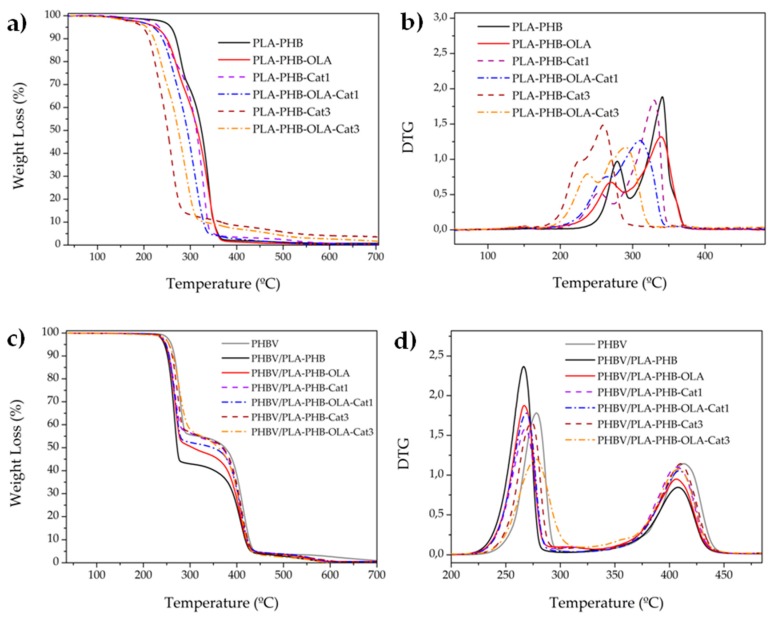
Dynamic TGA measurements of electrospun mats: (**a**) TGA and (**b**) DTG, as well as of bilayer systems: (**c**) TGA and (**d**) DTG.

**Figure 4 nanomaterials-09-00346-f004:**
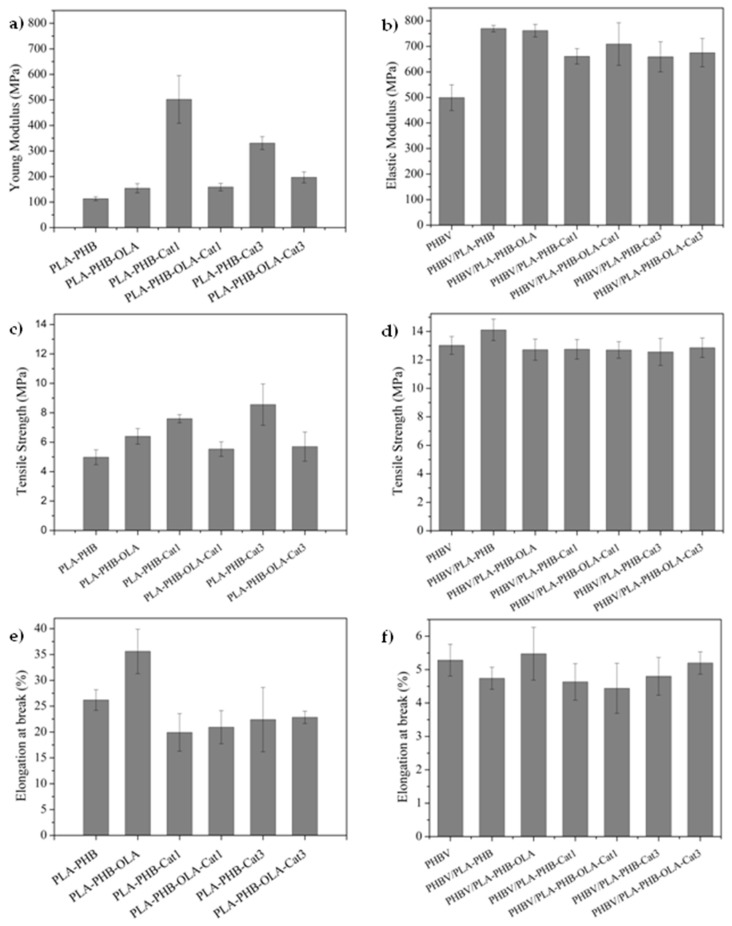
Mechanical properties of electrospun monolayer mats (**left**) and bilayer systems (**right**).

**Figure 5 nanomaterials-09-00346-f005:**
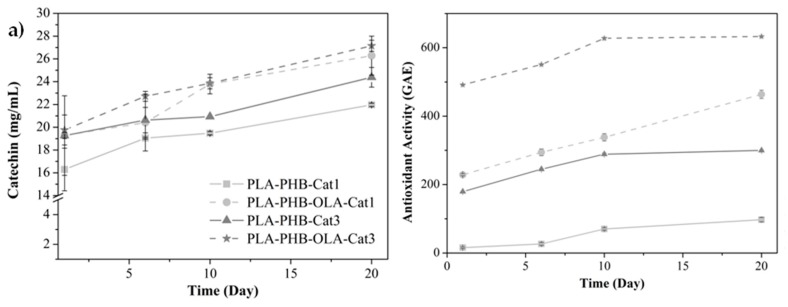
(**a**) Catechin release from bilayer materials to the food simulant and (**b**) antioxidant activity expressed as gallic acid concentration measured by DPPH radical scavengers.

**Figure 6 nanomaterials-09-00346-f006:**
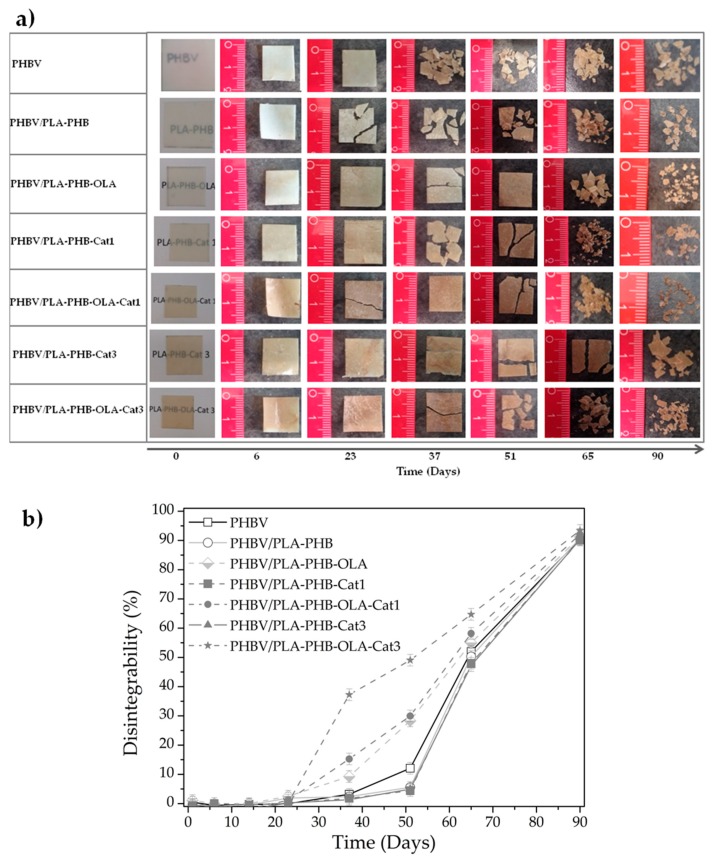
Disintegration under composting conditions: (**a**) Visual appearance of bilayer films before and after different incubation days under composting conditions, and (**b**) disintegration degree of bilayer films as a function of time under composting conditions.

**Table 1 nanomaterials-09-00346-t001:** Electrospun PLA-PHB fibers and their composites.

Formulations	PLA (wt%)	PHB (wt%)	OLA (wt%)	Cat (wt%)	Fibers Diameter (nm)	Dinamic Viscosity (η) (Pa.s)
PLA-PHB	75	25	-	-	215 ± 67	0.12 ± 0.01
PLA-PHB-OLA	63.75	21.25	15	-	260 ± 78	0.06 ± 0.01
PLA-PHB-Cat1	74.25	24.75	-	1	405 ± 143	0.13 ± 0.03
PLA-PHB-OLA-Cat1	63.0	21.0	15	1	228 ± 57	0.06 ± 0.02
PLA-PHB-Cat3	72.75	24.25	-	3	400 ± 116	0.14 ± 0.01
PLA-PHB-OLA-Cat3	61.5	20.5	15	3	206 ± 57	0.07 ± 0.01

**Table 2 nanomaterials-09-00346-t002:** TGA and DTG results of the electrospun fiber mats.

Electrospun Mats	T_0_ (°C)	T_max1_ (°C)	T_max2_ (°C)
PLA-PHB	258.1	278.7	341.0
PLA-PHB-OLA	225.0	270.2	339.2
PLA-PHB-Cat1	232.9	253.8	329.7
PLA-PHB-OLA-Cat1	220.3	263.2	310.4
PLA-PHB-Cat3	197.5	270.8	344.7
PLA-PHB-OLA-Cat3	205.9	250.8	282.5

**Table 3 nanomaterials-09-00346-t003:** TGA and DTG results of bilayer systems.

Electrospun Mats	T_0_ (°C)	T_max1_ (°C)	T_max2_ (°C)
PHVB	260.0	278.2	413.1
PHBV/PLA-PHB	244.9	266.6	407.8
PHBV/PLA-PHB-OLA	247.5	266.9	406.3
PHBV/PLA-PHB-Cat1	248.5	267.8	406.7
PHBV/PLA-PHB-OLA-Cat1	248.5	268.8	408.9
PHBV/PLA-PHB-Cat3	254.7	273.6	410.1
PHBV/PLA-PHB-OLA-Cat3	255.2	277.3	408.2

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
