# Peer review of "Antioxidant Bilayers Based on PHBV and Plasticized Electrospun PLA-PHB Fibers Encapsulating Catechin"

_nanomaterials, 2019, doi:10.3390/nano9030346_

Round 1

Reviewer 1 Report

This manuscript deals with the development of a biobased and biodegradable bilayer systems with antioxidant properties formed by a compression molded 
outer layer and an antioxidant electrospun inner layer. The structure of the manuscript and experiments should be improved. The proposed approach is very interesting however there no clear analysis of the compressed moulded final results in term of electrospun membrane. The SEM images are not clear enough to understand if the nanocomposite structure is maintained after assembly. It is not clear the reason of the use of such expensive technique as electrospinning, if the nanofiber structure and encapsulation of catechin are not maintained after compression moulding. The system should be compared with a chatechin-containing membrane prepared by casting. I recommend its publication after major revision.

Author Response

We thank reviewer 1 for the interest in the proposed approach of the manuscript, for his/her valuable comments as well as for considering our work for its publication in Nanomaterials special issue: "Nanomaterials to Enhance Food Quality, Safety, and Health Impact".

-Regarding the electrospun membrane morphology on the compressed molded system the reviewer is totally right. Therefore, in the current version of the manuscript, new SEM images of the inner surface of bilayer system PHBV/PLA-PHB-OLA obtained after the compressed molded treatment have been included (new Figure 1i and 1j). We believe that now these results have been clarified and, thus, the paper has been improved.

-Regarding the choice of electrospinning technique instead of solvent casting process, it should be mentioned that casting process is not scalable to the industrial level, while electrospinning technique currently is scalable to the industrial level. Nowadays, electrospinning technique is used to process several kinds of materials at industrial level.

See for example the following link:

https://bioinicia.com/electrospinning-electrospraying-technology/

Therefore, some comments regarding the electrospinning technique as well as the interest on the use of the electrospinning technique for food packaging purposes has been added in the introduction section of the manuscript, considering the up-to-date literature.

At this regard, we can underline also that the non-woven fiber structure is maintained in the bilayer system, as confirmed with the new figures added in the revision version of the manuscript (new Figure 1i and 1j), confirming the good choice to use electrospun fiber mats as innovative way to obtain active bi or multilayer systems.

Reviewer 2 Report

Reviewing the Manuscript ‘Antioxidant bilayers based on PHBV and plasticized electrospun PLA-PHB fibers encapsulation catechin’ (nanomaterials-448507) by Arrieta et al.

Arrieta et al. describe the preparation of biodegradable antioxidant bilayer systems for food contact. This is nowadays a hot topic and the manuscript contains relevant information. Nevertheless, in my opinion the manuscript needs to be carefully revised before being accepted for publication.

 My main doubt is related to the method of preparation of the bilayer. Why did the authors used compression moulding to prepare the PHBV layer? Why they did not use electrospinning? Or alternatively, why the authors did not use compression moulding to produce the PLA-PHB layer? The authors claim a good dispersion of catechin within the electrospun fibers, but when they prepare the bilayer system by compression moulding, the fibers are melted, right? So, in this case, how the authors can ensure a good dispersion of catechin? From my perspective, this does not make sense.

The authors also say that the plasticized systems containing catechin have an improved thermal stability due to its good dispersion in the matrix. How was this evaluated? It should be also noted that this is only true for the systems containing 3% catechin.

Regarding the outer layer, the authors say that it is PHBV, but the commercial product that it is used is a blend in which PBAT is in higher percentage. Why did the authors used this blend instead of others with higher percentage of PHBV or even pure PHBV? Please clarify.

Also, the authors use OLA to improve the ‘stretchability’ of the final mats, but the final bilayer system has an elongation at break considerably lower than the inner layer. My question is, why did the authors invested in the elongation of the inner layer? Is related with its ‘resistance’ during the compression moulding?

Author Response

Reviewing the manuscript ’Antioxidant bilayers based on PHBV and plasticized electrospun PLA-PHB fibers encapsulation catechin’ (nanomaterials-448507) by Arrieta et al. Arrieta et al. describe the preparation of biodegradable antioxidant bilayer systems for food contact. This is nowadays a hot topic and the manuscript contains relevant information. Nevertheless, in my opinion the manuscript needs to be carefully revised before being accepted for publication.

We thank Reviewer 2 for his/her interest in the manuscript as well as for his/her valuable comments. We have now carefully revised the manuscript.

 My main doubt is related to the method of preparation of the bilayer. Why did the authors used compression moulding to prepare the PHBV layer? Why they did not use electrospinning?

Or alternatively, why the authors did not use compression moulding to produce the PLA-PHB layer? The authors claim a good dispersion of catechin within the electrospun fibers, but when they prepare the bilayer system by compression moulding, the fibers are melted, right? So, in this case, how the authors can ensure a good dispersion of catechin? From my perspective, this does not make sense.

We thank reviewer questions. To clarify these points, in the current version of the manuscript the method of preparation was better described. In the bilayer system, the fibers were not melted and maintained their structure. The temperature used is to obtain the bilayer system is not enough to melt the fibers. In the current version of the manuscript, as it was suggested also by Reviewer 1, the structural aspect of the electrospun inner layer in the final bilayer system have been now studied by SEM, where it is possible to observed that the non-woven fiber structure is maintained in the final material.

The authors also say that the plasticized systems containing catechin have an improved thermal stability due to its good dispersion in the matrix. How was this evaluated? It should be also noted that this is only true for the systems containing 3% catechin.

Thank you for this observation. In order to clarify this point and to improve the thermal characterization, in the current version of the manuscript TGA and DTG figures have been included and deeply commented. Additionally, isothermal TGA analysis of electrospun inner layer has been included.

Regarding the outer layer, the authors say that it is PHBV, but the commercial product that it is used is a blend in which PBAT is in higher percentage. Why did the authors used this blend instead of others with higher percentage of PHBV or even pure PHBV? Please clarify.

Thank you for this observation. Since PHBV still shows some disadvantages in terms of mechanical performance, it is frequently compounded with aliphatic-co-aromatic biodegradable polyester, such as PBAT. In this work, we selected a commercial compound which is commercialized as PHBV, under the trade name ENMAT 5010P. This is why we designed the outer layer as PHBV. Nevertheless, in the current version of the manuscript, some comments on this fact have been added in the introduction section.

Also, the authors use OLA to improve the ‘stretchability’ of the final mats, but the final bilayer system has an elongation at break considerably lower than the inner layer. My question is, why did the authors invested in the elongation of the inner layer? Is related with its ‘resistance’ during the compression moulding?

Thank you for this observation. In fact, OLA was used not only to increase the stretchability of the fiber based material required for film manufacturing, but also to facilitate the electrospinning process. These aspects have been clarified in the abstract as well as in the introduction sections of the manuscript.

We hope that the current version of the manuscript sill be now suitable for publication in the Special Issue "Nanomaterials to Enhance Food Quality, Safety, and Health Impact" of the Journal Nanomaterials.

Looking forward to hearing from you

 Best regards

 Marina P. Arrieta, PhD

Round 2

Reviewer 1 Report

The author corrections have improved significantly the structure of the manuscript and the understanding of the results. For these reasons, I recommend the publication of the manuscript as full paper without additional modifications.

Reviewer 2 Report

The authors took into consideration all of my suugestions and the manuscript can be accepted in its present form.